# Inhibitory Effects of Cycloheterophyllin on Melanin Synthesis

**DOI:** 10.3390/molecules26092526

**Published:** 2021-04-26

**Authors:** Joong-Hyun Shim

**Affiliations:** Department of Cosmetic Science, Semyung University, Science & Engineering Bldg/Room 313, Chungbuk 390-711, Korea; jhshim@semyung.ac.kr; Tel.: +82-43-6491615; Fax: +82-43-649-1730

**Keywords:** cycloheterophyllin, melanin, tyrosinase, microphthalmia-associated transcription factor

## Abstract

This study was performed to clarify the inhibitory effects of cycloheterophyllin on melanin synthesis. In order to elucidate the inhibitory effects of cycloheterophyllin on the B16F10 cell line, cell viability, messenger ribonucleic acid (mRNA) expressions, tyrosinase activity assay, and melanin production assay were measured. The effects of cycloheterophyllin on tyrosinase-related protein 1 (*TYRP1*)/*TYRP2*/tyrosinase (*TYR*)/microphthalmia-associated transcription factor (*MITF*) mRNA expressions and melanin content were determined. Quantitative real-time RT-PCR showed that cycloheterophyllin decreased the mRNA expression level of *TYRP1/TYRP2/TYR/MITF* genes and melanin production contents than α-MSH-treated B16F10 cells. The tyrosinase activity assay revealed that cycloheterophyllin decreased the melanin production in the B16F10 cells. These data show that cycloheterophyllin increases the whitening effects in the B16F10 cells; thus, cycloheterophyllin is a potent ingredient for skin whitening. Thus, further research on the mechanism of action of cycloheterophyllin for the development of functional materials should be investigated.

## 1. Introduction

The color of the skin is determined by hemoglobin in the blood, melanin pigment in the epidermal layer, and β-carotene in the subcutaneous tissue. Among them, the color of the skin is mainly determined by the amount and distribution of the melanin pigment [1]. Melanin is a pigment synthesized from the melanosome in the melanocyte that is present in the basal layer. It absorbs or scatters ultraviolet (UV) rays and has a beneficial function to suppress damage to the skin cells or skin tissues. However, excessive accumulation of melanin may cause spots, freckles, and pigmentation. It is known to cause hyperpigmentation [2,3].

Melanin pigment is a representative irritant that causes exogenous aging. It is produced by the melanogenesis of melanocytes that have been exposed to UV rays, and has a beneficial function of protecting the skin from UV rays by darkening the skin color. The melanin pigment consists of eumelanin (black) and pheomelanin (red brown) [4,5]. Melanin is produced by using L-tyrosine, a type of amino acid, as a substrate, and through L-DOPA (3,4-dihydroxyphenylalanine) and further oxidized to DOPA-quinone by the main enzymes tyrosinase (TYR), tyrosinase related protein-1 (TYRP1), and TYRP2. The DOPA-quione is further oxidized to red-yellow pheomelanin in the presence of sulfhydryl groups. The DOPA-quinone also decarboxylated, oxidized, and catalyzed into eumelanin when sulfhydryl groups are depleted [4,5]. Tyrosianse enzyme is the key enzyme responsible for the production of melanin, and it is known that licorice extract, Japanese oak extract, arbutin, and kojic acid, which are representative whitening materials, inhibit the activity of tyrosinase enzyme [6,7]. In particular, arbutin plays a role in inhibiting the activity of tyrosinase by reacting competitively with L-tyrosine, which is a raw material for melanin, and kojic acid bound to the active site of the tyrosinase by Cu^2+^. Kojic acid and arbutin are known to have strong whitening effects, but side effects of safety issues such as skin irritation have been reported [8,9]. Many studies are being conducted to discover new effective materials such as the development of natural-derived functional materials [8,9].

Cycloheterophyllin(C_30_H_30_O_7_), a type of prenylflavone, is abundantly contained in *Artocarpus heterophyllus*, and has been reported to have pharmacological and biological functions including anti-inflammatory, anti-platelet activity, and antioxidant efficacy [10,11,12,13]. In addition, it has been reported that it has antioxidant and anti-aging effects of human dermal fibroblasts induced by UV-A [14,15].

As such, it has been reported that cycloheterophyllin is effective in various diseases, but there is no research on the effects of melanin production by cycloheterophyllin. In this study, I examined the effects of cycloheterophyllin on the expression of markers *TYRP1, TYRP2, TYR,* and *MITF* in B16F10 cells, a melanoma cell line derived from mice, related to the melanin production mechanism.

## 2. Results and Discussion

CCK-8 assay was performed to determine the cytotoxicity of cycloheterophyllin in the B16F10 cell line. Various concentrations of cycloheterophyllin were added to measure the cell viability (Figure 1). When treated with a concentration of 20 μg/mL or more, it was confirmed that the survival rate of B16F10 cells significantly decreased compared to the control group. When cycloheterophyllin was added with a concentration of 10 μg/mL, the survival rate of B16F10 cells was similar to that of the control group. Hence cycloheterophyllin was added at concentrations of 10, 1, and 0.1 μg/mL in additional experiments.

Tyrosinase enzyme reacts with L-tyrosine to produce L-DOPA, and L-DOPA finally produces melanin pigment through DOPAquinone by tyrosinase, which is known as a key enzyme that regulates melanogenesis [4,5]. In this experiment, kojic acid was used as a positive control to evaluate the degree of inhibition of tyrosinase enzyme activity. The degree of inhibition of tyrosinase enzyme activity was confirmed by the treatment with cycloheterophyllin at concentrations of 10, 1, and 0.1 μg/mL. It was confirmed that tyrosinase enzyme activity was inhibited in a concentration-dependent manner by cycloheterophyllin (Figure 2). In the cycloheterophyllin-treated groups at concentrations of 10, 1, and 0.1 μg/mL, it was confirmed that the inhibitory effects of tyrosinase enzyme were 58.0, 27.4, and 16.9%, respectively.

UV rays stimulate keratinocytes in the epidermis to induce the expression of α-melanocyte-stimulating hormone (α-MSH), a hormone that stimulates melanocytes, and is secreted in outside the cell, and the secreted α-MSH stimulates melanocytes. By binding to melanocortin 1 receptor (MC1R), it induces signaling related to melanin synthesis in the cell, causing the production of melanin pigment [5]. In this experiment, in order to induce melanin production in B16F10 cells, 200 nM α-MSH was used to induce melanin production in B16F10 cells [16]. Real-time RT-PCR of the expression levels of *TYRP1, TYRP2, TYR,* and *MITF* genes, which are markers that are increased by α-MSH treatment was determined, by cycloheterophyllin treatment in α-MSH-treated B16F10 cells. B16F10 cells were treated with cycloheterophyllin and the expression of the marker was confirmed through real-time RT-PCR. As a result, it was confirmed that the expressions of *TYRP1, TYRP2, TYR,* and *MITF* were significantly reduced in the cycloheterophyllin-treated group compared to the α-MSH-treated group. Specifically, in the experimental group treated with 10 μg/mL of cycloheterophyllin, the expression of *TYRP1, TYRP2, TYR*, and *MITF* genes decreased by 53.3, 26.6, 58, and 34%, respectively (Figure 3). In the experimental group treated with 1 μg/mL cycloheterophyllin, it was confirmed that the expression of the markers was significantly reduced, and the expression of *TYRP1* and *TYRP2* genes was similarly decreased in the 0.1 μg/mL cycloheterophyllin-treated group.

After confirming that the treatment of cycloheterophyllin reduced the mRNA expression level of a marker related to melanogenesis, the rate of melanin pigment production was confirmed to determine whether the production of melanin pigment in B16F10 cells was substantially reduced (Figure 4). B16F10 cell lines treated with cycloheterophyllin at concentrations of 10, 1, and 0.1 μg/mL, respectively, were harvested and the degree of melanin production was confirmed. As a result, it was confirmed that the production of melanin pigment was significantly reduced by cycloheterophyllin at concentrations of 10 and 1 μg/mL (Figure 4A). Additionally, it was confirmed that the production of melanin pigment in the cycloheterophyllin-treated condition was decreased compared to the control group (Figure 4B). When 10 and 1 μg/mL cycloheterophyllin was added, the production rate of melanin pigment decreased by 44.6 and 37.8%, respectively, which was significantly reduced compared to the control group (Figure 4B). These results have a tendency to be consistent with the experimental results of real-time RT-PCR on the mRNA expression levels of melanogenesis-associated marker.

Antioxidants such as ascorbic acid are known to be an important whitening material that inhibits the synthesis of melanin pigments by reducing DOPAquinone, an intermediate substance during the eumelanin production process, to L-DOPA [16,17]. Cycloheterophyllin has also been reported to have antioxidant effects [11,12]. It is shown that cycloheterophyllin may be effective in whitening by this antioxidant effect. As a result of confirming the antioxidant effect in vitro through this DPPH radical scavenging assay, it was confirmed that it has an antioxidant effect comparable to that of ascorbic acid, a positive control under high concentration of cycloheterophyllin (Figure 5). In the future, further research on the antioxidative effects of cycloheterophyllin in melanocytes and the expression of antioxidative enzymes such as catalase, SOD1, SOD2, and SOD3 should be conducted.

## 3. Materials and Methods

### 3.1. Experimental Materials and Cell Culture

B16F10 cell line was purchased from the Korean Cell Line Bank (Korea), and cell culture medium was 10% fetal bovine serum(FBS, SFBU−0500, Equitech-bio, Kerrville, TX, USA) in Dulbecco’s modified Eagle’s medium (DMEM, LM 001–05, Welgene, Gyeongsan-si, Korea), and 1% penicillin/streptomycin (Gibco, 15140–122, Waltham, MA, USA) was added. Cells were cultured in an incubator at 37 °C and 5% CO_2_ [16]. The cycloheterophyllin used in this experiment was extracted from *Artocarpus heterophyllus* using DMSO, and purchased from ChemFaces Co. (CFN97748, Wuhan, China, Purity ≥ 98%).

### 3.2. Cell Viability Assay

Cell viability was evaluated by CCK-8 solution (Cell counting kit; EZ−3000, EZ-Cytox, Seoul, DoGen). B16F10 cell lines were seeded at culture medium at a density 2 × 10^4^ cells/96 well plate. After 24 h cycloheterophyllin (CFN97748, Chemfaces) was added. B16F10 cell line treated with cycloheterophyllin were incubated for 24 h and CCK-8 solution was added for 30 min in a 37 °C incubator. Absorbance was measured at 450 nm using a microplate reader (Epoch, Winooski, VT, USA, BioTek) and cell viability was calculated based on the absorbance of the control group without cells [16].

### 3.3. DPPH Assay

For DPPH radical (Sigma-Aldrich, 382051, Darmstadt, Germany) scavenging assay, DPPH was dissolved in methanol:water (3:2) at 0.1 mM concentration. Diluted cycloheterophyllin (50 μL, CFN97748, Chemfaces) was added to DPPH solution (500 uL) and allowed to react at room temperature for 20 min, followed by a microplate spectrophotometer (Epoch, Winooski, VT, USA, BioTek,) and the absorbance was measured at a 517 nm wavelength. Ascorbic acid (1043003, Munich, Germany, Sigma-Aldrich) was used as a positive control.

### 3.4. In Vitro Mushroom Tyrosinase Activity

Mushroom tyrosinase (EC 1.14.18.1, Sigma-Aldrich) was dissolved in 1/15 M PBS (pH 6.8) to a concentration of 276 units/mL and used. About 150 μL of Mushroom Tyrosinase (13.8 units/mL) was reacted with L-tyrosine (T−3754, Sigma-Aldrich) and cycloheterophyllin (10, 1, 0.1 ug/mL) for 5 min, and the absorbance was measured at a wavelength of 475 nm. Each experimental group was independently tested three times, and the tyrosinase enzyme inhibition rate (%) of the material was calculated by applying the following calculation formula. As a positive control, kojic acid (K3125, Sigma-Aldrich) at a concentration of 100 μg/mL was used.

Tyrosinase inhibition ratio (%) = (1 − absorbance of treatment/absorbance of non-treatment) × 100

### 3.5. Quantitative Real-Time Reverse Transcription-Polymerase Chain Reaction (Q-RT-PCR)

Total RNA was isolated using Trizol (15596018, Carlsbad, CA, USA, Thermo Fisher Scientific). The concentration of RNA was determined spectrophotometrically (BioTek). Four micrograms of RNA was reverse-transcribed into cDNA using ReverTra Ace^®^ reverse transcription kit (Toyobo, FSQ101, Osaka, Japan). The reverse transcription was stopped by adding Tris–EDTA buffer (pH 8.0) to a total of 200 μL of cDNA solution. The TaqMan^®^ Gene Expression Assay sets were purchased from Applied Biosystems. Q-RT-PCRs were done according to the manufacturer’s instructions. Briefly, 20 μL of Q-PCR mixture contained 10 μL 2× TaqMan Universal Master Mix (Applied Biosystems, Foster City, CA, USA), 1 μL 20× Taqman expression assay (Applied Biosystems), and 50 ng cDNA. Q-PCR mixture was reacted in 95 °C for 20 s and 40 times cycling reaction (95 °C for 1 s and 60 °C for 20 s) with a StepOnePlus System (Applied Biosystems). The gene identification numbers for the TaqMan expression assay used in the Q-RT-PCR analyses are presented in Table 1. Human GAPDH (Applied Biosystems) was used for normalizing variation in cDNA quantities from different samples.

### 3.6. Melanin Contest Assay

To measure the amount of melanin pigment, Hosoi’s method was partially modified [17]. Total of 1 × 10^5^ B16F10 cells were inoculated into a 60 mm tissue culture dish. After culturing the cells for 24 h, 200 nM of α-melancyte stimulating hormone (α-MSH; M4135, Sigma-Aldrich) and cycloheterophyllin (10, 1, 0.1 μg/mL) were added for 72 h at each concentration. After removing the culture medium, 200 μL of NaOH (1N; S8045, Sigma-Aldrich) solution was added and incubated at 60 °C for 2 h to dissolve melanin. The absorbance was measured with a spectrophotometer (405 nm, Epoch). As a positive control, 100 μg/mL arbutin (A4256, Sigma-Aldrich) was used. Inhibition of the production of melanin pigment was expressed as a percentage of the amount of melanin produced under the α-MSH treatment conditions. It was performed by independently performing three independent experiments.

### 3.7. Statistical Analysis

Statistical analyses were carried out by the one-way analysis of variance (ANOVA). Results are expressed as the means ± standard deviation of at least three independent experiments.

## 4. Conclusions

UV encountered in daily life have beneficial functions such as vitamin D synthesis and sterilization, but in other aspects, it causes loss of skin elasticity, wrinkles, pigmentation, erythema, inflammation, etc., by a chain reaction of reactive oxygen species [18,19]. In addition, exogenous aging caused by external factors such as UV prevents the function of normal tissues and regeneration when the tissue is damaged due to the reduction and deterioration of the constituent cells in the living body [20,21]. 

Through this study, it was possible to confirm the appropriate concentration of cycloheterophyllin to treat the B16F10 cells. It was confirmed that the concentration of 10 μg/mL did not affect the viability of cells, as shown in Figure 1. By the measurement results of the activity inhibitory effect of the tyrosinase enzyme, it was confirmed that cycloheterophyllin concentrations of 10, 1, and 0.1 μg/mL significantly reduced the activity of the tyrosinase enzyme (Figure 2). In addition, it was confirmed that cycloheterophyllin significantly reduced the expression of markers (*TYRP1, TYRP2, TYR,* and *MITF* gene) compared to α-MSH treated condition (Figure 3). Additionally, it was confirmed that cycloheterophyllin reduced the amount of melanin synthesis in B16F10 cells (Figure 4). Based on these results, cycloheterophyllin shows the potential as a new whitening candidate.

This study is the first to confirm the inhibitory effect of cycloheterophyllin on melanin synthesis, and suggests the possibility of inhibiting skin aging caused by UV rays. In addition, further in-depth its effects on the signaling mechanisms of B16F10 cells, on the signaling systems involved in whitening, and on various antioxidant enzymes present in melanocytes seem to be necessary.

## Figures and Tables

**Figure 1 molecules-26-02526-f001:**
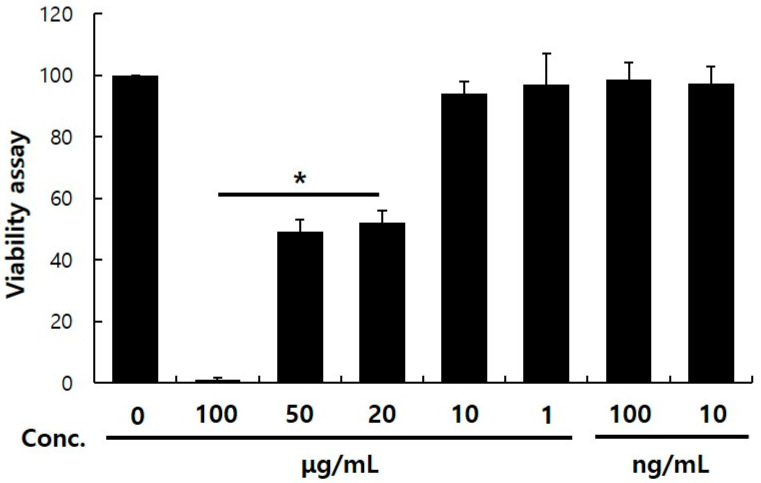
Cytotoxicity aspects of cycloheterophyllin on B16F10 cell line. B16F10 cells were seeded (2 × 10^4^ cells) in 96-well plate and treated with the indicated concentration of cycloheterophyllin for 24 h. Cell viability was measured by CCK-8 assay. Results are presented as the mean ± S.D. of the percentage of control optical density in triplicates. * *p* < 0.05 compared to control.

**Figure 2 molecules-26-02526-f002:**
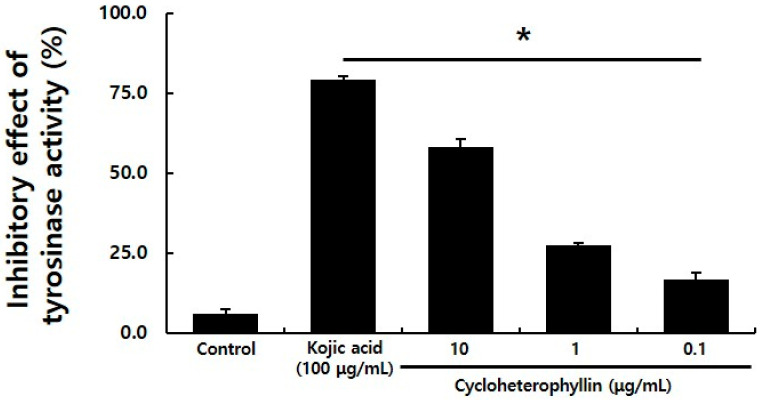
Tyrosinase activity inhibition effect of cycloheterophyllin. Results are the average of triplicate samples. * *p* < 0.05 compared with the control.

**Figure 3 molecules-26-02526-f003:**
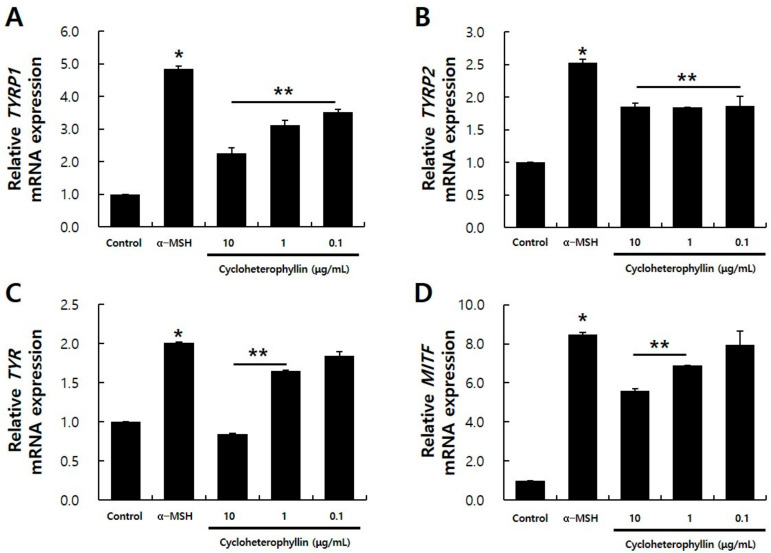
Characterization of cycloheterophyllin treatment on α-MSH-treated B16F10 cells. Real-time RT-PCR analysis of melanocyte markers *TYRP1* (**A**), *TYRP2* (**B**), *TYR* (**C**), and *MITF* (**D**). Values are mean ± S.D. of three independent experiments. * significantly different compared to control, *p* < 0.05 ** significantly different compared to α-MSH-treated condition, *p* < 0.05.

**Figure 4 molecules-26-02526-f004:**
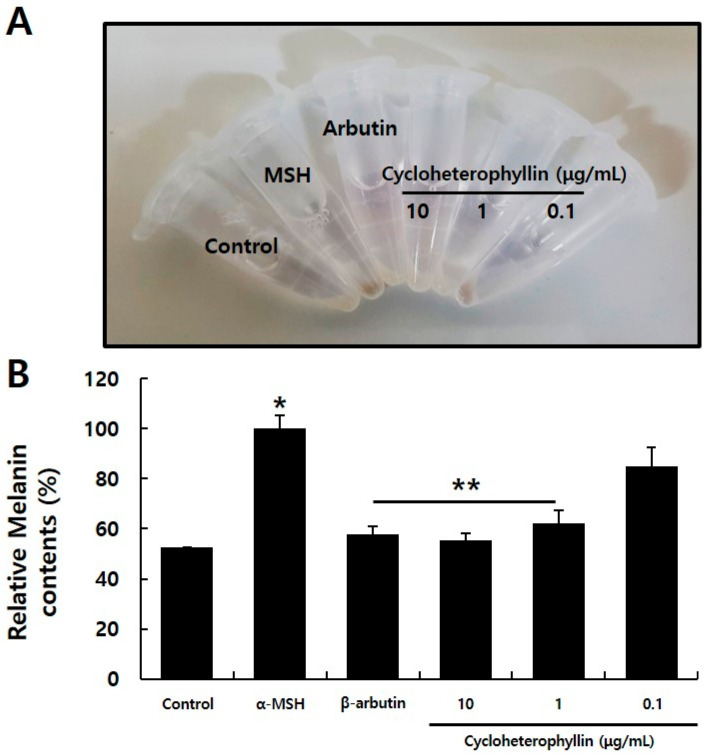
Melanin synthesis inhibition of cycloheterophyllin on B16F10 cells. Representative image of B16F10 cells after cycloheterophyllin treatment (**A**). Treated cells were lysed with 1 N NaOH and absorbance was measured at 405 nm (**B**). Results are expressed as means ± S.D. of three independent experiments. * compared to control, *p* < 0.05. ** compared to α-MSH-treated condition, *p* < 0.05.

**Figure 5 molecules-26-02526-f005:**
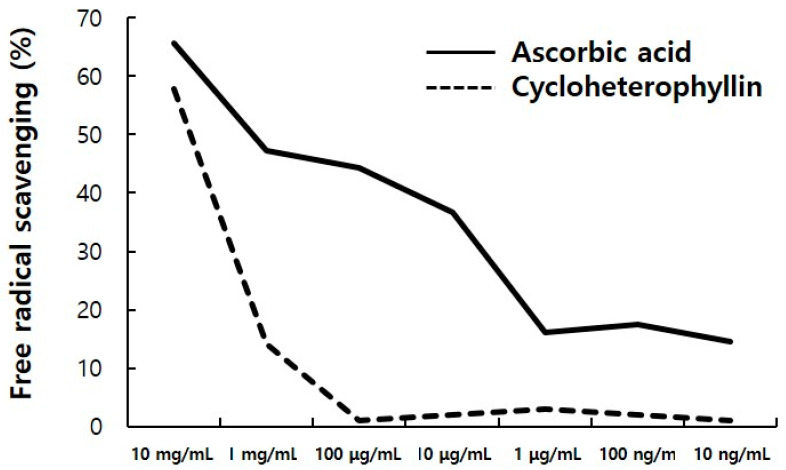
Free radical scavenging of cycloheterophyllin by the DPPH assay. The cycloheterophyllin has an antioxidant ability similar to ascorbic acid, positive control in high concentration of cycloheterophyllin.

**Table 1 molecules-26-02526-t001:** Gene name and assay ID number in real-time RT-PCR analysis.

Symbol	Gene Name	Assay ID
TYRP1	Tyrosinase-related protein 1	Mm00453201_m1
TYRP2	Tyrosinase-related protein 2	Mm01225584_m1
TYR	Tyrosinase	Mm00495817_m1
MITF	Microphthalmia-associated transcription factor	Mm00434954_m1
GAPDH	Glyceraldehyde-3-phosphate dehydrogenase	Mm99999915_g1

## Data Availability

The data presented in this study are available in article.

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
