# Peer review of "Inhibitory Effects of Cycloheterophyllin on Melanin Synthesis"

_molecules, 2021, doi:10.3390/molecules26092526_

Round 1
Reviewer 1 Report
The article is interesting. I have very few comments.
First, all Materials and methods section: all apparatus, accessories, software, reagents should also be well described in terms of their producers. It is not enough to mention only the country of production or only the name of the producer of the apparatus or reagent. All data must be provided, i.e. type of apparatus/hardware/software, model, name of reagent, manufacturer's name, city, state (e.g. in the case of the USA), country. There are many shortcomings in this regard in the reviewed article.
Figure 1A: The structure caption is unnecessary, because the figure caption explains what is in part A of the figure.
Author Response
First, all Materials and methods section: all apparatus, accessories, software, reagents should also be well described in terms of their producers. It is not enough to mention only the country of production or only the name of the producer of the apparatus or reagent. All data must be provided, i.e. type of apparatus/hardware/software, model, name of reagent, manufacturer's name, city, state (e.g. in the case of the USA), country. There are many shortcomings in this regard in the reviewed article.
Answer: I appreciate the reviewer’s close investigation on my manuscript. I corrected it according to reviewer`s instruction.
Figure 1A: The structure caption is unnecessary, because the figure caption explains what is in part A of the figure.
Answer: I thank for the thoughtful suggestion from the reviewer. I removed it in the figure 1A.

Reviewer 2 Report
The manuscript by Shim focuses on the inhibitory effects of cycloheterophyllin on melanin synthesis. Despite being quite clear, I have a few concerns about this article:
- Figure 1B: intermediary doses of cycloheterophyllin (20 and 50 ug/mL) should be used in order to get a dose-response curve
- Figure 3: Western Blot analysis of these makers should be performed to support RT-PCR data
- Figure 5: effects of cycloheterophyllin on ROS generation in B16F10 cells should be assessed (for instance, by colorimetric assay or flow cytometry with dichlorofluorescein)
- All figures: significancy (*) relative to C should be better indicated
Author Response
The manuscript by Shim focuses on the inhibitory effects of cycloheterophyllin on melanin synthesis. Despite being quite clear, I have a few concerns about this article:
Figure 1B: intermediary doses of cycloheterophyllin (20 and 50 ug/mL) should be used in order to get a dose-response curve
Answer: I thank for the thoughtful suggestion from the reviewer. I added new figure according to reviewer`s mention.
Figure 3: Western Blot analysis of these makers should be performed to support RT-PCR data
Answer: I appreciate the reviewer’s close investigation on my manuscript. The reviewer`s mention is right. But as shown in Fig. 4, the final result of Fig. 3 was confirmed by the amount of melanin produced, so what the reviewer said can be confirmed through Fig. 4. And it is realistically impossible to carry out the Western blot analysis in a short period of revision time (within 10 days). So I would be grateful if you could understand this situation.
Figure 5: effects of cycloheterophyllin on ROS generation in B16F10 cells should be assessed (for instance, by colorimetric assay or flow cytometry with dichlorofluorescein)
Answer: I thank for the thoughtful suggestion from the reviewer. It is realistically impossible to carry out the Western blot analysis in a short period of revision time (within 10 days). I think the comments of the reviewer could be another new story. And I`ll do further study and prepare new article about what the reviewer said. So I would be grateful if you could understand this situation.
All figures: significancy (*) relative to C should be better indicated
Answer: I appreciate the reviewer’s close investigation on my manuscript. In all figures, `* ` means the significance compared to control group. And in figure 3 and 4, ‘**` means the significance compared to a-MSH treated group. I specified these contents in figure legends.

Reviewer 3 Report
In the work “Inhibitory Effects of Cycloheterophyllin on Melanin Synthesis” the author studied the inhibitory effects of cycloheterophyllin on the B16F10 cell line. The author found that the cycloheterophyllin was able to reduce the activity of tyrosinase and melanin synthesis in a concentration-dependent manner. Despite the good results, the manuscript is very hard to follow and needs an extensive English revision to match the standards of Molecules. In its current state the manuscript is not suitable for publication.
Several questions need to be addressed:
Abstract section – what is the rationale beyond using the cycloheterophyllin for the development of healthy food?
Introduction section:
- “Among them, the color of the skin is mainly determined by the amount and distribution of the melanin pigment. Melanin, a brown or black pigment present in the skin, is known to be an important factor in determining the color of skin and hair [1].” – Please consider revising. The same information is repeated in both sentences.
- Please refer to melanin as a pigment and not as a protein.
- “After oxidation of L-DOPA, eumelanin and pheomelanin are finally produced through an additional mechanism [4,5].” – Please describe in more detail the mechanism of eumelanin and pheomelanin production.
- “Kojic acid and arbutin are known to have strong whitening effects, but side effects of safety issues such as skin irritation have been reported.” – Please add references
- “….there is no research on the mechanism of melanin production by cycloheterophyllin.” – Please revise the sentence. The cycloheterophyllin is not responsible for melanin production.
- “Through this, the possibility of cycloheterophyllin as a future whitening functional raw material and biomaterial is suggested.” – Please elucidate the potential of cycloheterophyllin as a biomaterial.
Materials and methods section:
- Please indicate the concentrations of cycloheterophyllin used in all assays.
- Cell viability assay - Did the author let the B16F10 cells adhere after seeding, and before adding the cycloheterophyllin?
- “In vitro Mushroom tyrosinase activity” – It is very hard to follow the protocol, please revise.
- “Quantitative real-time reverse transcription-polymerase chain reaction (Q-RT-PCR)” – Please indicate the conditions used for the assay. The compounds used to treat the sample?
- “Melanin contests assay” – should be Melanin content assay. Please revise the protocol since it is not clear how the author performed the assay.
Results and Discussion section:
- “CCK-8 assay was performed to determine the concentration of cycloheterophyllin” – CCK-8 assay was used to determine the cytotoxicity of cycloheterophyllin and not the concentration of cycloheteropphyllin. Please revise
- “When cycloheterophyllin was treated with a concentration of 10 μg/mL or less, the survival rate of B16F10 cells was similar to that of the control group.” – The B16F10 cells were treated with the cycloheterophyllin. Please revise
- “In additional experiments, cycloheterophyllin at concentrations of 10, 1, and 0.1 μg/mL were treated.” – please revise the sentence
- “In this experiment, kojic acid was used as a positive control to evaluate the degree of inhibition of tyrosinase enzyme activity of cycloheterophyllin, and the degree of inhibition of tyrosinase enzyme activity was confirmed by treatment with cycloheterophyllin at concentrations of 10, 1, and 0.1 μg/mL.” – Please revise the sentence. Is very long and hard to follow
- Figure 2 caption – please indicate what is the control
- “UV rays do not directly stimulate melanocytes to produce melanin, but produce melanin through detailed signal transduction in several stages.” – Please add references
- “In this experiment, in order to induce melanin production in B16F10 cells, 200 nM α-MSH was treated to induce melanin production in B16F10 cells [16].” – was used instead of was treated
Conclusions section:
- “Through this study, it was possible to confirm the appropriate concentration of cycloheterophyllin to be treated in B16F10 cells.” – to treat and not to be treated
Author Response
In the work “Inhibitory Effects of Cycloheterophyllin on Melanin Synthesis” the author studied the inhibitory effects of cycloheterophyllin on the B16F10 cell line. The author found that the cycloheterophyllin was able to reduce the activity of tyrosinase and melanin synthesis in a concentration-dependent manner. Despite the good results, the manuscript is very hard to follow and needs an extensive English revision to match the standards of Molecules. In its current state the manuscript is not suitable for publication.
Answer: I appreciate the reviewer’s close investigation on my manuscript. Before submission, I got the English revision service. If I got the final accept, I`ll additional English revision service from other English revision company.
Several questions need to be addressed:
Abstract section – what is the rationale beyond using the cycloheterophyllin for the development of healthy food?
Answer: I thank for the thoughtful suggestion from the reviewer. I deleted `development of healthy food` to avoid confusion.
Introduction section:
“Among them, the color of the skin is mainly determined by the amount and distribution of the melanin pigment. Melanin, a brown or black pigment present in the skin, is known to be an important factor in determining the color of skin and hair [1].” – Please consider revising. The same information is repeated in both sentences.
Answer: I appreciate the reviewer’s close investigation on my manuscript. As the review`s mention, I deleted one sentence.
Please refer to melanin as a pigment and not as a protein.
Answer: I thank for the thoughtful suggestion from the reviewer. I corrected it according to reviewer`s instruction.
“After oxidation of L-DOPA, eumelanin and pheomelanin are finally produced through an additional mechanism [4,5].” – Please describe in more detail the mechanism of eumelanin and pheomelanin production.
Answer: I appreciate the reviewer’s close investigation on my manuscript. I described more detail according to reviewer`s instruction.
“Kojic acid and arbutin are known to have strong whitening effects, but side effects of safety issues such as skin irritation have been reported.” – Please add references
Answer: I thank for the thoughtful suggestion from the reviewer. I added references.
“….there is no research on the mechanism of melanin production by cycloheterophyllin.” – Please revise the sentence. The cycloheterophyllin is not responsible for melanin production.
Answer: I appreciate the reviewer’s close investigation on my manuscript. I modified the sentece from “mechanism of ~“ to “effects of ~“.
“Through this, the possibility of cycloheterophyllin as a future whitening functional raw material and biomaterial is suggested.” – Please elucidate the potential of cycloheterophyllin as a biomaterial.
Answer: I thank for the thoughtful suggestion from the reviewer. I deleted the sentence because it seems to cause confusion.
Materials and methods section:
Please indicate the concentrations of cycloheterophyllin used in all assays.
Answer: I appreciate the reviewer’s close investigation on my manuscript. . I corrected it according to reviewer`s instruction.
Cell viability assay - Did the author let the B16F10 cells adhere after seeding, and before adding the cycloheterophyllin?
Answer: I thank for the thoughtful investigation from the reviewer. When I study CCK-8, cells were seeded B16F10. Afetr 24 hr, I treated cycloheterophyllin in the condition of adherent state. I corrected the sentences to get rid of confusion.
“In vitro Mushroom tyrosinase activity” – It is very hard to follow the protocol, please revise.
Answer: I appreciate the reviewer’s close investigation on my manuscript. I corrected it according to reviewer`s instruction.
“Quantitative real-time reverse transcription-polymerase chain reaction (Q-RT-PCR)” – Please indicate the conditions used for the assay. The compounds used to treat the sample?
Answer: I thank for the thoughtful investigation from the reviewer. I corrected it according to reviewer`s instruction.
“Melanin contests assay” – should be Melanin content assay. Please revise the protocol since it is not clear how the author performed the assay.
Answer: I thank for the thoughtful investigation from the reviewer. I corrected it according to reviewer`s instruction.
Results and Discussion section:
“CCK-8 assay was performed to determine the concentration of cycloheterophyllin” – CCK-8 assay was used to determine the cytotoxicity of cycloheterophyllin and not the concentration of cycloheteropphyllin. Please revise
Answer: I appreciate the reviewer’s close investigation on my manuscript. I corrected it according to reviewer`s instruction.
“When cycloheterophyllin was treated with a concentration of 10 μg/mL or less, the survival rate of B16F10 cells was similar to that of the control group.” – The B16F10 cells were treated with the cycloheterophyllin. Please revise
Answer: I thank for the thoughtful investigation from the reviewer. I corrected it according to reviewer`s instruction.
“In additional experiments, cycloheterophyllin at concentrations of 10, 1, and 0.1 μg/mL were treated.” – please revise the sentence
Answer: I appreciate the reviewer’s close investigation on my manuscript. I corrected it according to reviewer`s instruction.
“In this experiment, kojic acid was used as a positive control to evaluate the degree of inhibition of tyrosinase enzyme activity of cycloheterophyllin, and the degree of inhibition of tyrosinase enzyme activity was confirmed by treatment with cycloheterophyllin at concentrations of 10, 1, and 0.1 μg/mL.” – Please revise the sentence. Is very long and hard to follow
Answer: I thank for the thoughtful investigation from the reviewer. . I corrected it according to reviewer`s instruction.
Figure 2 caption – please indicate what is the control
Answer: I thank for the thoughtful suggestion from the reviewer. I modified the figure 2 because it seems to cause confusion.
“UV rays do not directly stimulate melanocytes to produce melanin, but produce melanin through detailed signal transduction in several stages.”
Answer: I appreciate the reviewer’s close investigation on my manuscript. I added reference.
“In this experiment, in order to induce melanin production in B16F10 cells, 200 nM α-MSH was treated to induce melanin production in B16F10 cells [16].” – was used instead of was treated
Answer: I thank for the thoughtful suggestion from the reviewer. I corrected it according to reviewer`s instruction.
Conclusions section:
“Through this study, it was possible to confirm the appropriate concentration of cycloheterophyllin to be treated in B16F10 cells.” – to treat and not to be treated
Answer: I appreciate the reviewer’s close investigation on my manuscript. I corrected it according to reviewer`s instruction.

Round 2
Reviewer 2 Report
The manuscript by Shim has not been significantly improved. However, it can be accepted as it is if the other reviewers do not have any more concern.
Author Response
Reviewer 2.
The manuscript by Shim has not been significantly improved. However, it can be accepted as it is if the other reviewers do not have any more concern.
Answer: I appreciate the reviewer’s close investigation on my manuscript.

Reviewer 3 Report
The manuscript was improved, however, there are still several questions that need to be addressed before acceptance.
Line 43 – “…dulfhydryl groups are delpleted.” Should be “sulfhydryl groups are depleted.”
Line 80 – Please add the cycloheterophyllin used in the experiment
Line 88 - Please add the cycloheterophyllin used in the experiment
Line 122 – “Melanin contests assay” should be “Melanin content assay”
Line 124 – 128 – “After culturing the cells for 24 hours, 200 nM of α-melancyte stimulating hormone (α-MSH; M4135, 125 Sigma-Aldrich) and cycloheterophyllin (10, 1, 0.1 μg/ml) were treated for 72 hours at each concentration. After removing the culture medium, 200 μl of NaOH (1N; S8045, Sigma- Aldrich) solution was treated to and dissolved melanin at 60 °C for 2 hours” - please revise since it is confusing and with some mistakes
Line 141 – “at various concentrations” should be “with various concentrations”
Line 144 – 145 – The cells were treated with cycloheterophyllin and not the cycloheterophyllin was treated. Please revise
Line 145 – “than 10 ug/ml” – please remove
Line 145 – If the cycloheterophyllin did not affect the cellular viability for concentration equal to or lower than 10 μg/ml why did the authors tested the cytotoxicity of 0, 1, and 0.1 μg/ml concentrations in an additional experiment?
Figure 1 – Why Figure 1A and 1B? Except for 20 and 50 μg/ml the data is already present in Fig 1A. Please revise since it is very confusing.
Line 170 - UV rays do not directly stimulate melanocytes to produce melanin, but produce melanin through detailed signal transduction in several stages [5].” – Please remove the sentence.
Author Response
Reviewer 3.
The manuscript was improved, however, there are still several questions that need to be addressed before acceptance.
Line 43 – “…dulfhydryl groups are delpleted.” Should be “sulfhydryl groups are depleted.”
Answer: I appreciate the reviewer’s close investigation on my manuscript. I corrected it according to reviewer`s instruction.
Line 80 – Please add the cycloheterophyllin used in the experiment
Line 88 - Please add the cycloheterophyllin used in the experiment
Answer: I thank for the thoughtful suggestion from the reviewer. I added the information.
Line 122 – “Melanin contests assay” should be “Melanin content assay”
Answer: I appreciate the reviewer’s close investigation on my manuscript. I corrected it according to reviewer`s instruction.
Line 124 – 128 – “After culturing the cells for 24 hours, 200 nM of α-melancyte stimulating hormone (α-MSH; M4135, 125 Sigma-Aldrich) and cycloheterophyllin (10, 1, 0.1 μg/ml) were treated for 72 hours at each concentration. After removing the culture medium, 200 μl of NaOH (1N; S8045, Sigma- Aldrich) solution was treated to and dissolved melanin at 60 °C for 2 hours” - please revise since it is confusing and with some mistakes
Answer: I thank for the thoughtful suggestion from the reviewer. I corrected it according to reviewer`s instruction.
Line 141 – “at various concentrations” should be “with various concentrations”
Answer: I appreciate the reviewer’s close investigation on my manuscript. I corrected it according to reviewer`s instruction.
Line 144 – 145 – The cells were treated with cycloheterophyllin and not the cycloheterophyllin was treated. Please revise
Answer: I thank for the thoughtful suggestion from the reviewer. I corrected it according to reviewer`s instruction.
Line 145 – “than 10 ug/ml” – please remove
Answer: I thank for the thoughtful suggestion from the reviewer. I deleted the sentence because it seems to cause confusion.
Line 145 – If the cycloheterophyllin did not affect the cellular viability for concentration equal to or lower than 10 μg/ml why did the authors tested the cytotoxicity of 0, 1, and 0.1 μg/ml concentrations in an additional experiment?
Answer: I appreciate the reviewer’s close investigation on my manuscript. When I treated 10 ug/ml or lower than 10 ug/ml, cell viability of B16F10 was similar to control group. This is because if the cells die, the experiment is impossible, and if the concentration is too low, it is difficult to check the effect of the material. In order to find the concentration that can exert the effect of the material without dying the cells, I determined the concentration of 10, 1, and 0.1 ug/ml in additional experiments.
Figure 1 – Why Figure 1A and 1B? Except for 20 and 50 μg/ml the data is already present in Fig 1A. Please revise since it is very confusing.
Answer: I thank for the thoughtful suggestion from the reviewer. It was a request from another reviewer, so I modified it. I`ll revise the figure and text according to reviewer`s instruction.
Line 170 - UV rays do not directly stimulate melanocytes to produce melanin, but produce melanin through detailed signal transduction in several stages [5].” – Please remove the sentence.
Answer: I appreciate the reviewer’s close investigation on my manuscript. . I corrected it according to reviewer`s instruction.
